# ZJUSAH Classification: A New Classification for Primary Brainstem Hemorrhage

**DOI:** 10.3390/life13030846

**Published:** 2023-03-21

**Authors:** Jingyi Zhou, An Ping, Jizhong Mao, Yichen Gu, Fengqiang Liu, Anwen Shao

**Affiliations:** 1Department of Neurosurgery, Second Affiliated Hospital, School of Medicine, Zhejiang University, Hangzhou 310009, China; 2Brain Research Institute, Zhejiang University, Hangzhou 310058, China; 3Collaborative Innovation Center for Brain Science, Zhejiang University, Hangzhou 310058, China; 4Clinical Research Center for Neurological Diseases of Zhejiang Province, Hangzhou 310009, China

**Keywords:** primary brainstem hemorrhage, state of consciousness, multiple logistic regression, predictive factors

## Abstract

To analyze and improve ZJUSAH classification for primary brainstem hematoma, we retrospectively reviewed 211 patients with primary brainstem hemorrhage who were admitted to our institution between January 2014 and October 2020. The primary clinical outcomes were the 30-day survival rate and 90-day consciousness recovery rate, which were evaluated using the National Institutes of Health Stroke Scale score. Univariate logistic regression and multivariate Cox regression analyses were performed to evaluate the prognostic model. The overall 30-day survival rate of the 211 patients was 69.7%. The 30-day survival rate was 95% among Type 1 patients, 77.8% among Type 2 patients, and 63.2% among Type 3 patients. The 90-day consciousness recovery rate was 63.2% among Type 1 patients, 61.9% among Type 2 patients, and 30.2% among Type 3 patients. Our findings suggest that ZJUSAH classification can be optimized according to hematoma volume, with Type 3 patients with a hematoma larger than 12.4 mL tending to have a worse state of consciousness. Additionally, we discovered that ZJUSAH classification is valuable in predicting 30-day survival rates in conservative treatment patients. In conclusion, our study established and optimized a new CT-based hematoma classification system for primary brainstem hematoma, which facilitates treatment selection and prognostic prediction.

## 1. Introduction

Primary brainstem hemorrhage (PBH) is a severe type of intracerebral hemorrhage (ICH) that is associated with hypertension, which poses major challenges for management [1,2,3,4,5] and prognostic prediction [6,7,8,9,10,11,12,13]. Brainstem hemorrhage caused by arteriovenous malformation, cerebral cavernous angioma, basilar tip aneurysms, brainstem glioma, or trauma are not considered to be PBH [14]. Due to the high number of PBH cases occurring in the pontine, some studies have used the term primary pontine hemorrhage (PPH) [13,15], which usually encompasses situations where the tegmentum or midbrain is involved as well. In this work, however, PBH will be used to refer to all primary hemorrhages in the brainstem.

PBH can cause brain injury through two mechanisms: the primary mass effect and the secondary cytotoxic effect of the hematoma [16]. These effects result in an increase in intracranial pressure, thus leading to reduced cerebral perfusion and potentially causing brain herniation [17]. Brainstem hematomas extending to the medulla can cause a disruption in respiratory and circulatory functions [13], while the extension to the midbrain may affect the reticular system, inducing coma [18]. Previous studies have demonstrated that the severity of neurological symptoms and the presence of hydrocephalus at the onset are strong prognostic indicators of poor outcomes [19,20,21,22,23,24]. PBH patients usually display rapidly progressing neurological manifestations [3].

Currently, the mainstream treatments for PBH are conservative treatments that focus on intensively monitoring neurological symptoms [25]. However, several surgical techniques—including open craniotomy, stereotactic precise aspiration, endoscopic surgery, and external ventricular drainage (EVD)—have been proposed as treatments for PBH (for review, see [6]), though a lack of clinical evidence has rendered them controversial [15,26]. Theoretically, surgical removal of the hematoma could reduce intracranial pressure, as well as attenuate the inflammatory effect and the parenchymal damage caused by the hematoma [16]. Minimally invasive techniques such as stereotactic precise aspiration and endoscopic surgery can minimize the surgical perturbation by entering through safe entry zones [27]. For instance, lesions in the ventral brainstem are always challenging to deal with due to the presence of numerous motor tracts [27]. Minimally invasive surgeries offer advantages for accessing this area, as they can be done through the supratrigeminal/peritrigeminal zone and suprafacial/infrafacial zone [6], thus minimizing the disruption of the surrounding tissue.

Since the 1980s [28,29], stereotactic precise aspiration has been used as a minimally invasive approach for treating cerebral hemorrhage. A retrospective study [30] compared this approach to conventional treatment for severe PBH and reported better outcomes in patients who underwent surgical treatment. Due to the advantages of less trauma, shorter surgical duration, and faster postoperative recovery, stereotactic precise aspiration has become a hotspot for the treatment of PBH [6].

The rapid diagnosis of PBH is critical due to the rapid progression of the disease [3]. While computed tomography (CT) and magnetic resonance imaging (MRI) have been found to have equivalent accuracy in diagnosing acute ICH [31], CT is still preferred, as it is more cost-effective and time-saving [3]. Thus, CT is the first option for diagnosing acute PBH [3]. Existing classifications of PBH hematomas are mainly based on their radiographic features, including volume and extension.

In 1986, Russel et al. [22] classified PBH into three distinct types: the central type, the dorsolateral tegmental type, and the tegmentobasilar type. Hematomas caused by hypertension are usually located at the center of the pontine, leading to rapid and lethal clinical progression [22]. The latter two types are also known as partial pontine hematomas, and they are generally caused by vascular malformations. Chung et al. [32] reported 62 cases of PBH and classified them into four types: the massive type (involving bilateral basal pontine and tegmentum), the bilateral tegmental type (involving bilateral tegmentum), the basal-tegmental type (involving the boundary of bilateral basal pontine and tegmentum), and the small unilateral tegmental type (involving only unilateral tegmentum). However, Fong et al. [21] conducted a study (n = 39) and found that these types may not be ideal for prognostic prediction. Wessels et al. [19] proposed a classification system that classified PBH into dorsal, ventral, and massive types after analyzing 29 patients’ clinical data. Studies that used this classification suggested that the ventral and massive types are associated with higher mortality rates, while the dorsal type is associated with a better prognosis. Although previously proposed classifications have varied, they have all noted a correlation between larger hematoma volumes and radiographic evidence of hydrocephalus and poor prognosis [19,20,21,22,23,24]. Unfortunately, no consensus has been reached on PBH classification, making it difficult to predict survival outcome and functional recovery.

Based on our previous findings [33], we discovered that the volume and location (primarily involving the brainstem) of a hematoma were significantly associated with the 30-day survival outcomes and 90-day consciousness recovery rates. Consequently, CT scans were essential for localizing the hematoma before stereotactic aspiration, thus leading to the introduction of ZJUSAH classification. This classification considers both the volume and location of the hematoma at the same time. Our goal was to take this further by studying and assessing clinical characteristics and outcomes in classified patients and evaluating the performance of ZJUSAH classification for prognostic prediction. This study aimed to collect and analyze clinical data from PBH patients to identify the prognostic factors and develop a prognostic prediction model. Additionally, the study also sought to investigate and optimize the new brainstem hematoma classification method: ZJUSAH classification, which is a more straightforward approach for the prognostic prediction of PBH based on cranial CT evidence collected upon emergency administration.

## 2. Methods

### 2.1. Study Design

This single-center, retrospective study was conducted with the approval of the ethics committee of our institution. The STROBE guidelines [34] for observational studies were followed in the present study. Due to the observational nature of this study and the harmlessness to patients, the need for informed consent was waived. During follow-ups, the objectives of this study were explained to patients or their families.

### 2.2. Patient Selection

We retrospectively reviewed the clinical data of 211 patients with PBH who were admitted to our institution between January 2014 and October 2020. The inclusion criteria were as follows: (1) available cranial CT and CT angiography (CTA) results within 24 h of admission; (2) available records of laboratory test results, vital signs, and GCS score at admission; (3) patients were followed up with to assess clinical outcomes. The exclusion criteria included secondary brainstem hematoma caused by trauma, intravenous thrombolysis, cavernous malformation, arteriovenous malformation, or tumors.

The patients received either conservative treatments or stereotactic surgeries during hospitalization. Indications for stereotactic surgeries included: (1) a hematoma volume ≥ 5 mL and a GCS score ≤ 7; (2) a hematoma volume of 3~5 mL and a GCS score ≤ 5, with conservative treatment for at least 72 h from disease onset; (3) PBH accompanied by acute hydrocephalus. Surgical contradictions included a GCS score ≤ 3, bilateral mydriasis, unstable vital signs, or other signs of brainstem function failure.

### 2.3. Clinical Data

All patients’ clinical data were reviewed, including demographic characteristics, clinical characteristics at admission (vital signs and GCS scores), laboratory test results and radiological results (cranial CT and CTA) at admission or during hospitalization, and complications that occurred during hospitalization. Hematoma volume was determined in the (A × B × C)/2 manner, where A is the greatest hematoma diameter by CT, B is the diameter 90° to A, and C is the approximate number of CT slices with hematoma multiplied by the slice thickness [35].

### 2.4. Outcome Assessment

Primary outcomes included short-term survival (30-day survival rate) and mid-to-long-term recovery of consciousness (90-day consciousness recovery rate). Outcomes were obtained from patients or their families through telephone interviews conducted by two trained neurosurgeons who were blinded to the research data. The interviews were conducted 3 months after the patient’s discharge. The 90-day consciousness recovery rates were evaluated according to the National Institutes of Health Stroke Scale (NIHSS) score, with 0, 1, or 2 for consciousness and 3 for unconsciousness.

### 2.5. Prognostic Analysis Based on ZJUSAH Classification

Based on the hematoma location and the clinical requirements for stereotactic surgeries, we developed a four-type system for PBH classification (Figure 1A): Type 0, the hematoma is restricted to the cistern or the fourth ventricle, with the brainstem compressed but undamaged; Type 1, the hematoma is located within one side of the midline without any involvement of the other side; Type 2, the hematoma spans both sides of the midline, with or without the involvement of the right or left quarter of the brainstem; Type 3, the hematoma spans both sides of the midline, involving both the right and left quarter of the brainstem.

Furthermore, based on the relative location of the hematoma body to the ventral-dorsal diameter in the pons (Figure 1), Type 2 and 3 hematomas can be further classified into three subtypes: ventral type (A), dorsal type (B), and central type (C). For example, if the main body of a hematoma (more than 50%) is located in the ventral part of the brainstem, then it should be classified as ventral type (A) regardless of where the remaining parts are located. Note that Figure 1A only shows a limited number of possible situations. For more schematic illustrations of Type 2 and Type 3 hematomas, see Appendix A.

Here, we provided an efficient approach to estimate the right/left quarter borderline of the brainstem in CT images, as illustrated in Figure 1B: First, identify the intersection of the ventral–dorsal midline (marked in green) and the tangent line of the ventrolateral edge (marked in yellow). Then, draw lines (marked in red) starting from the intersection, equally dividing the angles in half. Representative CT images for each type of ZJUSAH classification are presented below (Figure 2).

### 2.6. Univariate Analysis and Survival Analyses Based on ZJUSAH Classification

R (version 4.0.3) and RStudio (version 1.4.1106) were used for statistical calculations. To evaluate the predictive performance of ZJUSAH classification, inter-group comparisons of outcomes were conducted using one-way ANOVA tests. The equality of variance was determined using Levene’s test. If the one-way ANOVA identified significant differences, univariate logistic regression was performed to evaluate the predictive performance of ZJUSAH classification on the outcomes.

Type 1, Type 2, and Type 3 patients were included in subsequent analyses (one case of Type 0 patient was excluded). A Kaplan–Meier survival curve was plotted for 30-day survival outcomes, and the statistical significance was evaluated using the log-rank test. Multivariate survival analysis (of ZJUSAH Classification and treatments) was performed using Cox regression models. Hazard ratios (HRs) and 95% confidence intervals (CIs) were calculated for each variable. The probability of *p* < 0.05 was considered to be statistically significant.

## 3. Results

### 3.1. Prognostic Analyses of Primary Brainstem Hematoma Patients Based on ZJUSAH Classification

A total of 211 patients were included in the study. There was 1 case of Type 0, 20 cases of Type 1, 54 cases of Type 2 (3 Type 2A cases, 19 Type 2B cases, and 32 Type 2C cases), and 136 cases of Type 3 (7 Type 3A cases, 68 Type 3B cases, and 61 Type 3C cases).

The overall 30-day survival rate of the 211 patients was 69.7%. The only Type 0 case died in the hospital after conservative treatment. Type 1 patients had a 30-day survival rate of 95% (19/20). Type 2 patients had a 30-day survival rate of 77.8% (42/54), with 63.2% (12/19) for Type 2A, 63.2% (12/19) for Type 2B, and 87.5% (28/32) for Type 2C. Type 3 patients had a 30-day survival rate of 63.2% (86/136), with 57.1% (4/7) for Type 3A, 55.9% (38/68) for Type 3B, and 72.1% (44/61) for Type 3C (Table 1).

An intra-group analysis was conducted to compare the effect of conservative treatment and stereotactic surgery on Type 1, Type 2, and Type 3 patients, respectively. Among Type 3 patients, a significant difference in 30-day survival rates was observed between the two modalities: The 30-day survival rate for the stereotactic surgery group was 76.9%, while the survival rate for the conventional treatment group was 44.8% (*p* < 0.05).

A total of 208 patients were followed up with for the 90-day consciousness recovery rate, with 3 patients lost to follow-up (1 Type 1 patient and 2 Type 2C patients). The overall 90-day consciousness recovery rate was 63.2% (12/19) among Type 1 patients. Type 2 patients had a consciousness recovery rate of 61.9% (26/42), with 66.7% (2/2) for Type 2A, 26.3% (5/12) for Type 2B, and 63.3% (19/28) for Type 2C. Type 3 patients had a consciousness recovery rate of 30.2% (26/86), with 50.0% (2/4) for Type 3A, 18.4% (7/38) for Type 3B, and 38.6% (17/44) for Type 3C (Table 2).

An intra-group analysis was also conducted to compare the effect of 2 treatment modalities on Type 1, Type 2, and Type 3 patients. However, no significant difference was observed between the two modalities in any of the three groups (*p* > 0.05).

### 3.2. Optimization of ZJUSAH Classification Based on Hematoma Volume

To optimize the sensitivity of our classification to the volume of the hematoma (Figure 3), we established thresholds for hematoma volume in Type 2 and Type 3 patients. The average hematoma volumes in Type 2 and Type 3 patients were 7.5 ml (maximum = 20.6 mL, minimum = 0.9 mL) and 12.4 mL (maximum = 25.4 mL, minimum = 1.55 mL), respectively. These values were then used as the thresholds for grouping the patients into 2 groups for each type (Type 2 ≥ 7.5 mL and <7.5 mL; Type 3 ≥ 12.4 mL and <12.4 mL). Prognostic differences were calculated between the two groups. For Type 2 patients, no significant differences in 30-day survival rates or 90-day consciousness recovery rates were observed (Table 3). In contrast, significant differences in 90-day consciousness recovery rates were found between the two groups of Type 3 patients (*p* = 0.009). However, no significant differences in 30-day survival rates were observed (Table 4).

### 3.3. Survival Analysis for ZJUSAH Classification and Treatment Strategies

A Cox regression analysis was conducted to examine the effects of ZJUSAH classification and treatment modalities on 30-day survival outcomes. Type 1 was used as a reference, and the results indicated that Type 2A, Type 2C, and Type 3C had no significant differences compared to Type 1, whereas Type 2B, Type 3A, and Type 3B were associated with worse 30-day survival outcomes. Moreover, stereotactic aspiration was found to be protective, as it was associated with improved 30-day survival outcomes (HR = 0.28, 95% CI [0.16–0.47], *p* < 0.001) (Figure 4).

By analyzing survival curves for different types of ZJUSAH classifications, it was clear that there was a significant difference in survival outcomes between the three types of patients (*p* = 0.0087). Notably, patients receiving conservative treatment had a 30-day survival rate lower than those who received stereotactic aspiration treatment (*p* < 0.0001) (Figure 5).

### 3.4. Using ZJUSAH Classification to Predict 30-Day Survival Outcomes in Patients Receiving Conservative Treatments

The predictive effect of ZJUSAH classification was initially evaluated in all patients. However, the result (AUC = 0.676, 95% CI [0.60–0.75]) was not satisfactory. To further investigate, we analyzed the predictive effect of ZJUSAH classification on the conservative treatment group and found that it could successfully predict 30-day survival rates. Notably, dorsal brainstem hematomas (Type 2B and Type 3B) were significantly associated with a poorer prognosis (Figure 6).

## 4. Discussion

PBH is a highly lethal disease with a mortality ranging from 30–60% during its acute phase [36,37,38], making a rapid approach to identification essential. The invention of CT [39] revolutionized clinical practice and quickly became the primary diagnostic imaging approach for PBH due to its speed and informational value [20].

### 4.1. Comparison of Different Classifications

Since the 1980s, the PBH-related features of CT images have been widely discussed and studied [40]. Russel et al. [22] proposed a classification method for PBH that classified brainstem hematomas into central pontine hematoma, dorsolateral tegmental hematoma, and tegmentobasilar hematoma. In 1992, Chuang et al. [32] proposed another CT-based classification method, classifying brainstem hematomas into the massive type, the bilateral tegmental type, the basal-tegmental type, and the small unilateral tegmental type, with survival rates of 7.1%, 14.3%, 26.1%, 94.1%, respectively. Subsequent studies further improved upon Chuang’s classification [23,41]. A 2004 study by Wessels et al. [19] suggested that dorsal brainstem hematomas have a better prognosis than ventral brainstem hematomas. Thus, over the past 30 years, nearly all studies have agreed that unilateral tegmental hematomas have a relatively better prognosis, while massive hematomas have the worst prognosis. Patients between the two types often have varying survival rates and functional recovery. As a result, clinical symptoms and other representative lab assays should also be considered in the diagnostic panel of PBH to achieve better prognostic prediction. Recently, Kim et al. [42] proposed a method of using standard grids to evaluate the volume and location of the brainstem hematoma. Their retrospective study indicated that the initial hematoma volume was the most important factor for prognosis, while hematoma location has no relationship with neurological outcomes.

We proposed a new classification for brainstem hematoma in 2009 based on the name of our institution, namely the ZJUSAH classification. To facilitate rapid assessment in clinical or emergent situations where a prompt diagnosis is crucial, we classified PBH into Type 0, Type 1, Type 2, and Type 3 based on the location of the hematoma. Note that Type 0 hematomas located in the cistern/ventricle may not always invade the parenchyma of the brainstem; they can still cause symptoms due to compression. The prognostic model based on this classification guided us to find that the location and volume of brainstem hematoma are important factors affecting the short-term survival prognosis and mid-to-long-term recovery. Our classification quantifies the scope of the brainstem damage, which reflects the position of the hematoma in the brainstem, meanwhile reflecting the volume of the hematoma.

However, hematomas are classified according to their maximum cross-sections, and the volume characteristics of hematomas cannot be covered entirely. Therefore, we further optimized Type 2 and Type 3 in the ZJUSAH classification system and discovered that Type 2 patients with hematoma larger than 10 mL have a worse mid-to-long-term state of consciousness than those with smaller hematoma. Similarly, Type 3 patients with larger hematomas (larger than 12 mL) have worse mid-to-long-term states of consciousness than the ones with smaller hematomas. However, there were no differences in short-term survival outcomes in these comparisons.

Our study revealed a worse prognosis in dorsal subtypes of brainstem hematoma (Type 2B and Type 3B) than in non-dorsal subtypes (Type 2A, Type 3A, Type 2C, and Type 3C). This was especially true for the 90-day consciousness recovery rate, which was 41.7% for Type 2B vs. 61.9% for Type 2 and 18.4% for Type 3B vs. 30.2% for Type 3. It seems that our results contradict Wessels et al.’s. Those detrimental effects (to consciousness recovery) of Type B hematoma may be explained by damage to the reticulum activating system. Moreover, the involvement of important nuclei, such as raphe nuclei (serotonergic), the red nucleus (motor modulation), and substantia nigra (dopaminergic) can cause other possible detrimental effects [43].

### 4.2. Surgical Management of PBH

The success of brainstem surgery requires accurate selection of a “safe entry zone”, proper surgical approach, and appropriate surgical techniques. Safe entry zones are chosen to minimize the disruption of important nuclei or fibers. However, the selection of a safe entry zone requires the surgeon to have in-depth neuroanatomical knowledge of the brainstem and its vascular structures. Yang et al. [44] did an evidence-based analysis of common safe entry zones for brainstem surgeries: for hematoma in the pons, there are supratrigeminal/peritrigeminal zones and suprafacial/infrafacial zones; for hematoma in the midbrain, there are supracollicular/infracollicular zones; for hematoma in the medulla, the posterior median sulcus or lateral medulla are better safe entry zones. However, although surgeons always operate under the general principle of limiting disruptions during brainstem surgeries, delicate maneuvers are indispensable during surgeries. Since small vessels, such as perforating arteries, cannot be properly observed prior to the surgery, intraoperative exposure and observation of the parent vessel are essential [45]. Therefore, apart from preoperative imaging, the intraoperative consideration of the local blood supply is equally important for the safety of a PBH surgery.

One of the most debated points about the surgical management of ICH is the inevitable damage caused by surgical procedures [46]. Stereotactic aspiration surgeries have been based on the preoperative 3D reconstruction of the hematoma and the brainstem, and in some published cases [47], 3D-printed navigation has been used to achieve relatively good outcomes. The surgical approach and safe entry zone should be designed according to the distribution of the hematoma, avoiding important fiber tracts or nuclei. Imaging techniques, such as diffusion tensor imaging and diffusion tensor tractography, have been applied for the visualization of white matter tracts in brain stem surgeries, such as brainstem glioma [48] and brainstem cavernomas [49], which have enhanced the safety and accuracy of the surgeries. During the surgery, the aspiration was conducted through a small hole 4 mm in diameter. The hematoma was discharged in small amounts while rinsing the residual cavity using saline. No retraction or electrocoagulation occurred during the operation, limiting iatrogenic brainstem damage to the lowest level. During the operation, the patient’s vitals were intensively monitored in case of rare adverse events, including bradycardia or cardiac arrest.

Apart from stereotactic aspiration, endoscopic brainstem hematoma evacuation has increasingly been reported as a promising technique for PBH [50,51]. With the radiological information of the hematoma, a suitable approach through the safe entry zone is chosen to enter the brainstem and evacuate the hematoma, which usually includes the transclival approach (endoscopic endonasal transclival approach, or EETA) and the retrosigmoid approach [6]. However, controlling iatrogenic injury still poses great challenges, and it is recommended that it should be undertaken by experienced neurosurgeons, paying attention to the operating principles of “no traction of the brainstem, light suction of the hematoma, and weak electric coagulation of the responsible vascular” [3].

Another widely used surgical intervention for brainstem hemorrhage is EVD, which is usually applied to patients with evidence of hydrocephalus [52]. Early drainage can lower the incidence of cerebral fluid circulation disorder [53]. However, Murata et al. [54] found that external ventricular drainage did not lead to a significant clinical benefit in brainstem hemorrhage patients with hydrocephalus. In a single-center, retrospective study, Luney et al. [55] compared the efficacy of craniotomy and decompression, conservative treatment, and EVD in posterior fossa hemorrhage (with or without brainstem hemorrhage) and found that the incidence of hydrocephalus and intraventricular hemorrhage were significantly higher in the EVD treatment group.

### 4.3. Limitations

First, the data used in this retrospective study was obtained at a single institute, so our classification may not accurately represent the general population. Second, the lack of Type 0 and some other subtypes (Type 2A and Type 3A) is a shortcoming of this study. Due to the small sample size of Type 0 (n = 1), it was excluded from the survival analyses conducted in this study. Future studies that include a greater number of Type 0 patients will be conducted to incorporate them into our predictive model. We anticipate that a larger collection of clinical cases will help to further refine the ZJUSAH classification system and its ability to accurately predict outcomes. Third, due to the limitations of sample size and this being a single-center study, we did not include information regarding whether the patients received EVD treatment before or after the operation. In follow-up studies, we will expand the sample size and do a more detailed subgroup analysis of EVD, hematoma volume, and admission GCS based on the ZJUISAH classification. Moreover, although we chose the 90th day as the endpoint of evaluating the state of consciousness, follow-ups longer than 90 days should also be considered in further studies to evaluate long-term outcomes.

A truly comprehensive treatment for PBH requires the continuous evaluation and exploration of the morbidity, mortality, clinical outcomes, prognostic factors, clinical procedures, and possible surgical indications. We hope our study can provide evidence for the management of PBH. We also expect that further studies with larger clinical sample sizes and better quality will enhance PBH management.

## 5. Conclusions

The ZJUSAH classification system for primary brainstem hematomas provides a model to predict the 90-day consciousness outcomes of the most extensive (Type 3) hematoma based on its volume. Additionally, our model allows for a 30-day survival outcome assessment when conservative treatment is applied. This CT-based hematoma classification system has the potential to enable more precise treatment selection and prognostic prediction for PBH patients.

## Figures and Tables

**Figure 1 life-13-00846-f001:**
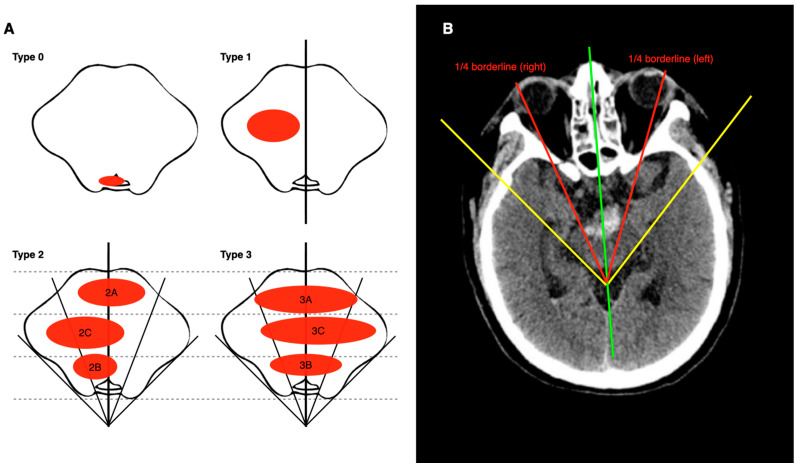
(**A**) Schematic representations of ZJUSAH classifications showing horizontal cross-sections of the brainstem are shown in the figure, with the fourth ventricle at the bottom. The dashed lines divide the ventral–dorsal diameter into three equal parts. (**B**) An easy way to find left and right 1/4 borderlines: The green line represents the ventral–dorsal midline. The yellow lines represent the tangent lines of the ventrolateral edges. The red lines represent the left and right 1/4 borderlines, starting from the intersection of green lines and yellow lines, equally dividing the angles in half.

**Figure 2 life-13-00846-f002:**
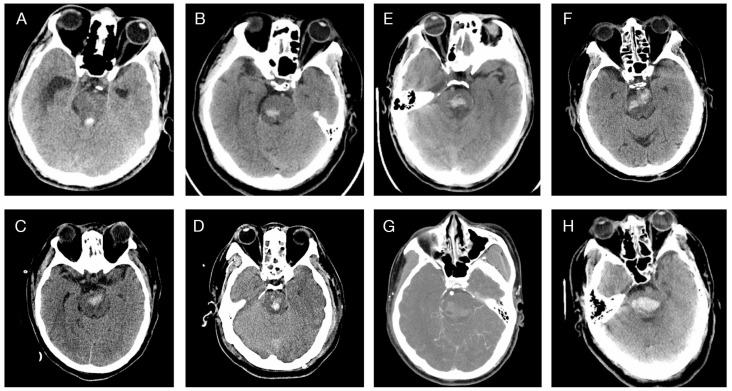
Representative cases of ZJUSAH classification. (**A**) The hematoma is within the cistern or the fourth ventricle, with the brainstem being compressed but not damaged (Type 0). (**B**) The hematoma remains on the right side of the brainstem without affecting the left side (Type 1). (**C**) The hematoma spans both the left and right sides of the midline, without involving the right or left 1/4 of the brainstem, localizing mainly within the ventral part of the brainstem (Type 2A). (**D**) The hematoma spans both sides, without involving the right or left 1/4 of the brainstem, localizing mainly within the dorsal part of the brainstem (Type 2B). (**E**) The hematoma spans both sides, without involving the right or left 1/4 of the brainstem, localizing mainly within the central part of the brainstem (Type 2C). (**F**) The hematoma spans both sides, involving 1/4 borderlines of both sides, localizing mainly within the ventral part of the brainstem (Type 3A). (**G**) The hematoma spans both sides, involving 1/4 borderlines of both sides, localizing mainly within the dorsal part of the brainstem (Type 3B). (**H**) The hematoma spans both sides, involving 1/4 borderlines of both sides, localizing mainly within the central part of the brainstem (Type 3C).

**Figure 3 life-13-00846-f003:**
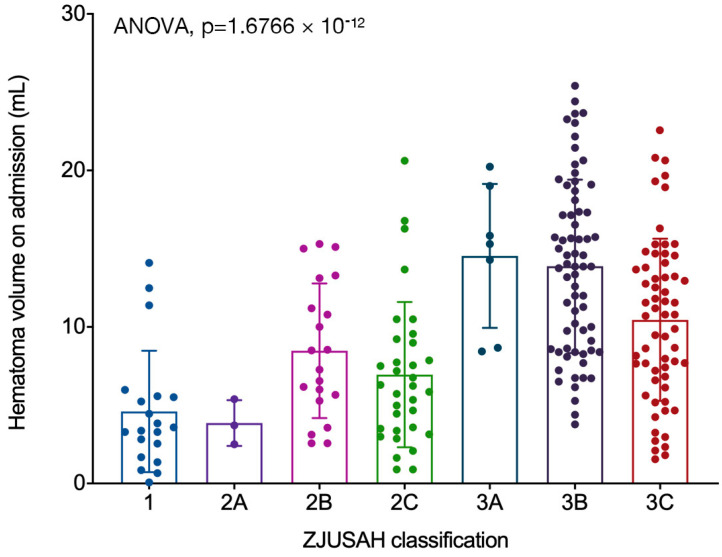
Distribution of the hematoma volumes of patients classified through ZJUSAH classification. Hematoma volumes in 210 Type 1, Type 2, and Type 3 patients. The error bar indicates the standard error of the mean (SEM).

**Figure 4 life-13-00846-f004:**
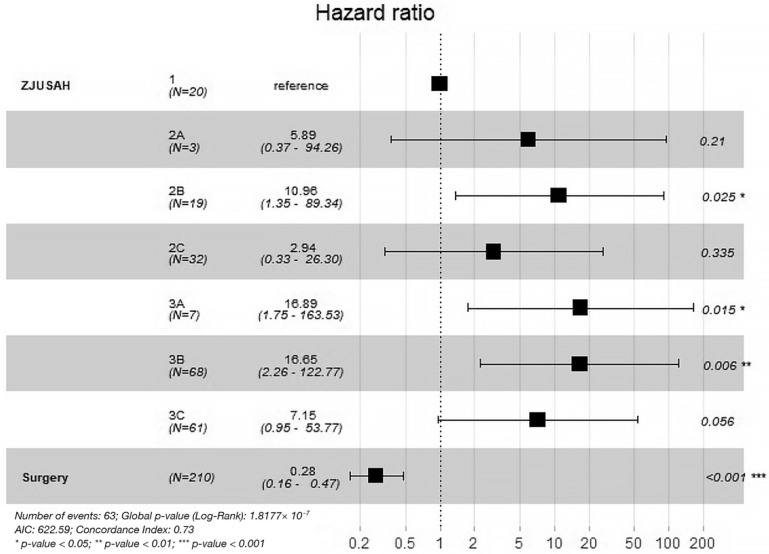
The 30-day death HR for different ZJUSAH classifications and different treatments. ZJUSAH stands for ZJUSAH classification; surgery stands for stereotactic aspiration surgeries.

**Figure 5 life-13-00846-f005:**
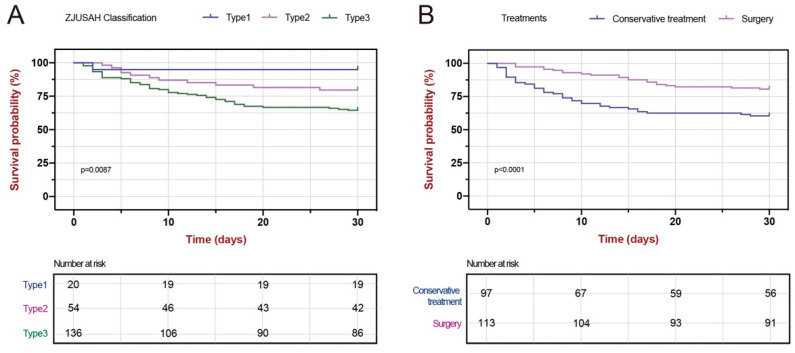
Kaplan–Meier Curve of different ZJUSAH classifications and different treatments. (**A**) Kaplan–Meier curve of 30-day survival stratified by ZJUSAH classification types. (**B**) Kaplan–Meier curve of 30-day survival stratified by treatments.

**Figure 6 life-13-00846-f006:**
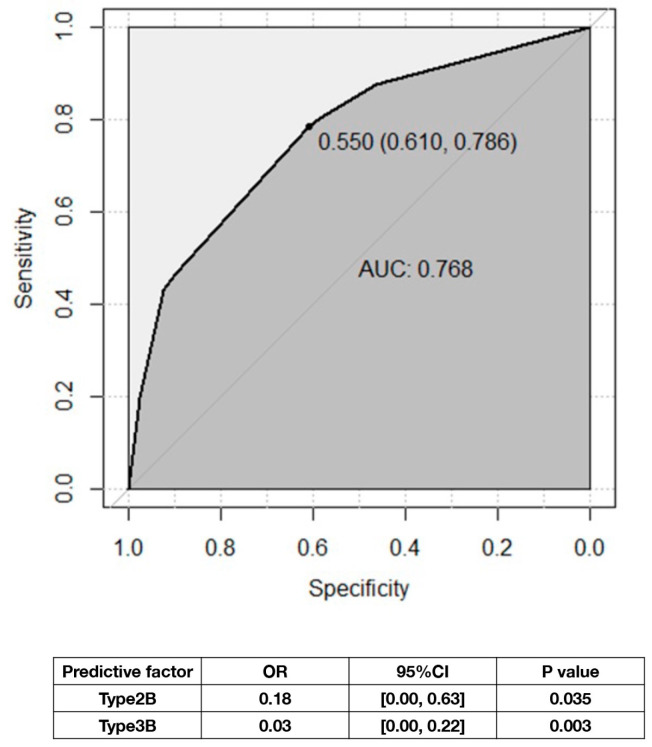
ROC curve: univariable logistic regression model of 30-day survival rate, ZJUSAH classification, and the predictive factors. Note that when Type 1 is set as the reference, Type 2B and 3B are unfavorable factors for 30-day survival outcomes, while the rest of the classifications show no significant correlation (*p* > 0.05).

**Table 1 life-13-00846-t001:** The 30-day survival outcomes for different ZJUSAH classifications.

ZJUSAH Classification	Patients in Total	Survivors (at Day 30)	30-Day Survival Rate (%)	*p*-Value
Type 0	1	0	0.0	0.004787
Type 1	20	19	95.0	
Type 2	54	42	77.8	
Type 2A	3	2	66.7	
Type 2B	19	12	63.2	
Type 2C	32	28	87.5	
Type 3	136	86	63.2	
Type 3A	7	4	57.1	
Type 3B	68	38	55.9	
Type 3C	61	44	72.1	
All patients	211	147	69.7	

**Table 2 life-13-00846-t002:** The 90-day consciousness recovery rate for different ZJUSAH classifications.

ZJUSAH Classification	Patients in Total	Conscious Patients (NIHSS Score < 3)	90-Day Consciousness Recovery Rate (%)	*p*-Value
Type 0	1	0	0.0	<0.001
Type 1	19	12	63.2	
Type 2	42	26	61.9	
Type 2A	2	2	100	
Type 2B	12	5	41.7	
Type 2C	28	19	67.9	
Type 3	86	26	30.2	
Type 3A	4	2	50	
Type 3B	38	7	18.4	
Type 3C	44	17	38.6	
All patients	148	40	27.0	

**Table 3 life-13-00846-t003:** Prognosis of Type 2 patients.

30-Day Survival Rate			
	≥7.5 mL	<7.5 mL	*p*
**Type 2**	**73.9% (17/23)**	**80.6% (25/31)**	**0.742**
Type 2A	NA	66.7% (2/3)	NA
Type 2B	60.0% (6/10)	66.7% (6/9)	1
Type 2C	84.6% (11/13)	89.5% (17/19)	1
**90-Day Consciousness Recovery Rate**			
	**≥7.5 mL**	**<7.5 mL**	** *p* **
**Type 2**	**30.4% (7/17)**	**61.3% (19/25)**	**1**
Type 2A	NA	100% (2/2)	NA
Type 2B	16.7% (1/6)	66.7% (4/6)	0.588
Type 2C	54.5% (6/11)	76.5% (13/17)	0.409

**Table 4 life-13-00846-t004:** Prognosis of Type 3 patients.

30-Day Survival Rate			
	≥12.4 mL	<12.4 mL	*p*
**Type 3**	**61.9% (39/68)**	**65.4% (47/68)**	**0.213**
Type 3A	40.0% (2/5)	100% (2/2)	0.429
Type 3B	50.0% (20/40)	64.3% (18/28)	0.322
Type 3C	73.9% (17/23)	71.1% (27/38)	1
**90-Day Consciousness Recovery Rate**			
	**≥12.4 mL**	**<12.4 mL**	** *p* **
**Type 3**	**15.4% (6/39)**	**42.6% (20/47)**	**0.009**
Type 3A	50.0% (1/2)	50.0% (1/2)	1
Type 3B	15.0% (3/20)	22.2% (4/18)	0.687
Type 3C	11.8% (2/17)	55.6% (15/27)	0.005

## Data Availability

Not applicable.

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
