# Peer review of "ZJUSAH Classification: A New Classification for Primary Brainstem Hemorrhage"

_life, 2023, doi:10.3390/life13030846_

Round 1

Reviewer 1 Report

Thank you for reporting this paper.

The title of the article is interesting to readers. The report was well written. however, there are some concerns that should be addressed:

- The authors mentioned in the abstract that "we ret-18 retrospectively reviewed 211 patients with primary brainstem hemorrhage admitted to our institution 19 from January 2014 to October 2020", but in the method section, they wrote that "we performed a retrospective analysis of patients with PBH who were administrated 109 between October 2010 and January 2014". hence, please modify the inclusion time precisely. I think this is a critical flaw that should be cleared.

- When did you do an interview with the patient? how many days/ months after discharge?

- please add limitations of the study after discussion. 

- please rewrite the conclusion. 

Reviewer 2 Report

The authors developed a novel radiological classification for primary brainstem haemorrhage aiming at prognostic prediction. Although other classifications exist, the authors argued their shortcomings and hence their attempt for improvement. It is an important study. This study included a relatively good number with 211 cases investigated. The classification is readily applicable and there is a clear definition and the method of application is adequately explained. The grades were correlated with mortality at 30 days and "wakefulness" at 90 days. The analysis was meticulous and with the optimisation by including the haematoma volume. As such there was correlation with outcome including the group managed conservatively.

To design such study aiming at assessing the impact of the classification on prognosis, ideally the design should: [1] include only those cases managed without intervention, [2] analyse other factors potentially impacting outcome some of which are important such as volume (which they added), GCS at presentation (not clear how analysed as there is very brief mention that this only correlated with the 90 days "wakefulness"), AGE and hydrocephalus, and finally [3] compare with other existing classification systems.

However, 113 had surgical intervention - stereotactic aspiration. In fact the decision for surgery was based upon GCS at presentation or later, volume or presence of hydrocephalus - i.e. not influenced by their classification system. A correlation of outcome with surgical stereotactic aspiration vs. conservative management was presented though. It was not clear whether there were cases subjected to CSF drainage ?(external ventricular drains) either alone or in combination with stereotactic aspiration. 

Either the authors present the analysis on the conservatively treated cases alone (there will be less numbers included) and include correlation with other factors including GCS, age and hydrocephalus and possibly compare to other classification systems OR if data not readily available or the numbers would compromise the analysis then to discuss the reasons for their current study design and inclusion of surgically treated cases and whether this influenced the conclusions or not.

However, if there is a good reason that the study was designed as such with the inclusions and the ommission of the other factors from the analysis then, this can be clearly stated in the manuscript. 

It will be nice to state what ZJUSAH stands for.  

Reviewer 3 Report

Dear authors, thank you for sharing your experience with us.

Regarding the retrospective study by Zhou et collegues, entitled ZJUSAH CLASSIFICATION: A NEW CLASSIFICATION FOR PRIMARY BRAINSTEM HEMORRAGE, please find below my comments

          Major revisions:

1)     In the opening part, some association to other possible etiologies? (ie coagulopathies, brainstem gliomas, basilar tip aneurysms, or even traumatic forms)

2)     Since the minimally invasive role of stereotactic surgery (targeted drainage of hemorrhage in the shortest time and affecting fewer neighboring structures) has been rightly emphasized: a brief digression regarding neuronavigation in preoperative planning, a mention of DTI and tractography

3)     Delving into the concept of minimally invasive, is there any role for the endoscopic technique?

4)     When should conservative treatment be considered useful compared with minimally invasive stereotactic treatment?

5)     In Wessels' classification (2004), ventral forms are defined as having a worse prognosis than dorsal forms; in your study, a worse prognosis is found in dorsal Type2B and Type3B forms. In looking for some other anatomical explanation besides the RAS ("Those detrimental effects of Type B hematoma may be explained by damage to the reticulum activating system") you might mention the major gray nuclei present in the pons, midbrain and medulla

Minor revisions:

6)     When you say "surgical removal of the hematoma not only reduces intracranial pressure, but also attenuates the inflammatory effect and parenchymal damage caused by the hematoma" you attribute a role still fundamental to neurosurgery today. After highlighting the efficacy of the stereotactic procedure, it might be useful to compare it with open-craniotomy maneuvers by describing the main "safe entry zones" for evacuation of exophytic hematomas surfacing at the pons ventral/dorsal surfaces?  (i.e. peritrigeminal zone, median sulcus above facial colliculus, acustic area, infra-facial zone or others)

7)     Any role for the pathological anatomy, i.e. Duret's flame hemorrhages?

8)     Taking into account the delicate anatomical area, it is certain that delicate surgical maneuvers are indispensable!: after the correct clarification of "no retraction or electrocoagulation occurred during the operation, limiting iatrogenic brainstem damage to the lowest level,"  might it be helpful to describe the main procedures that can be used in the hemostasis phases during and at the end of surgery? What instruments are you used to shatter the clot without determining too much traction of the brainstem?

9) please consider 10.3389/fnana.2021.675313 for the surgical strategy

Reviewer 4 Report

The authors reviewed a series of 211 patients with primary brainstem hemorrhages and they proposed a novel CT-based classification taking into consideration the volume and location of the hematoma. They evaluated the efficacy of this classification as a prognostic predictor of the 30-day survival and 90-day state of consciousness. 

In summary, the objective of this study was to analyze and improve the ZJUSAH classification for primary brainstem hematoma and to evaluate the efficacy of this classification as a prognostic predictor of the 30-day survival and 90-day state of consciousness. They concluded that they established a new CT-based hematoma classification (ZJUSAH Classification) for primary brainstem hematoma, which plays an instructive role in the treatment and prognostic judgment. I congratulate the authors on this interesting topic. However, there were multiple issues with the paper that could use substantial improvement. Also, there is a lack of novelty in this topic.

Abstract: 

* What does “ZJUSAH” Stands for?

Introduction:

* The introduction is appropriate to the subject, and the literature review is relevant.

* The author mentioned "PBH". Would you please spell the abbreviations when first mentioned?

* The author mentioned many sentences without references. Would you please add a reference for the following sentences?

1- Since a large proportion of PBH occur in the pontine, some studies used the term primary pontine hemorrhage, or PPH.

2- Brainstem hematoma extending to the medulla can cause disorder in respiratory or circulatory functions.

3- Brainstem hematoma extending to the midbrain can affect the reticulum system, causing coma.

4- PBH patients usually present with rapidly worsening neurological symptoms.

5- Currently, the mainstream treatment for PBH are conservative treatments aiming to intensively monitor the neurological symptoms.

6- Due to the advantages of less trauma, shorter surgical duration, and faster postoperative recovery, stereotactic precise aspiration has become a hotspot for the treatment of PBH.

7- PBH requires rapid diagnosis due to the rapid disease progression.

8- CT is superior to MRI considering the popularity and the duration of examination.

9- CT has been the first option for the diagnosis of acute PBH.

10- Hematoma caused by hypertension usually locate at the center of the pontine, leading to rapid and lethal clinical progression.

11- Studies used Wessels’s classification indicated that the ventral type and massive type related to higher mortality, while the dorsal type indicated better prognosis.

Methodology:

* Which guideline was used in writing this manuscript? Retrospective studies usually use STROBE guidelines.

* The author mentioned, "Patients were followed up until clinical outcomes." What do the authors mean by this sentence?

* The author mentioned, "Critical exclusion criteria were secondary brainstem hematoma caused by trauma, intravenous thrombolysis, cavernous malformation, arteriovenous malformation, etc..." What do the authors mean by etc? What does this include?

* The author mentioned, "mid-to-long-term state of consciousness (90-day consciousness rate)." What do the authors mean by consciousness rate? 

* It is better to include mRS scores in these patients as this is the most commonly used way of assessment of the patients.

* The authors mentioned that they use the NIHSS score but I could not find this in the results.

* The author mentioned, "Hemorrhage volume was determined in the (A × B × C)/2 manner, where A is the greatest hemorrhage diameter by CT, B is the diameter 90° to A, and C is the approximate number of CT slices with hemorrhage multiplied by the slice thickness." Would you please write a reference to this sentence?

* Why are the authors adding Type 0 classification to their classification that doesn’t include any brainstem hemorrhages?

* Why the authors are not including the craniocaudal hematoma extension in their classification?

Results:

* Why type 0 was excluded from the analysis?

* The authors mentioned on several occasions in the results section that the results are not shown here. Is there any explanation for that?

Discussion:

* Please divide the discussion section into subtitles.

* There is no novelty in your study as there a recent publication by Kim et al was explaining a similar classification system.

Kim M, Cho S, Hoon You S, Park J, et al. A Prognostic Model of Pontine Hemorrhage Based on Hemorrhage Volume and Location. Journal of Neurointensive Care 2021; 4(1): 21-29. DOI: https://doi.org/10.32587/jnic.2021.00339

* Would you please add a limitation section to your manuscript in the discussion section?

* The are too many sentences that are missing references to identify individually in this review. Would you please add a reference to any sentence in the discussion section that is not the author's own work?

General:

The level of the English language is poor and there are too many errors to identify individually in this revision. Hence, a revision by a professional is highly recommended.

Round 2

Reviewer 1 Report

Thank you for addressing the comments. 

Good job

Author Response

We really appreciate your feedback!

Reviewer 4 Report

Thank you for addressing most of my comments.

I have the following minor comments.

Methodology:

* WOULD YOU PLEASE MENTION IN THE METHODS SECTIONS IN THE MANUSCRIPT THAT YOU USED THE STROBE GUIDELINES IN WRITING THE MANUSCRIPT AND REFERENCE THE ORIGINAL STOBE ARTICLE?

* OLD COMMENT: Why are the authors adding Type 0 classification to their classification that doesn’t include any brainstem hemorrhages?

This is a very good question. Indeed Type 0 classification seems not having any brainstem hemorrhage according to our description: “Type 0, the hematoma is restricted to the cistern or the fourth ventricle, with the brainstem compressed but undamaged”. Despite the fact that a hematoma in the cistern/ventricle may not always invade the parenchyma of the brainstem, it may still cause symptoms by compressing it. To facilitate rapid assessment in clinical and emergent situations, we propose including Type 0 into our classification system. Further studies with more of Type 0 patients will be conducted to incorporate Type 0 into our predictive model.

NEW COMMENT: Would you please add this to your discussion?

Results: 

* Why type 0 was excluded from the analysis?

Because the sample size of type 0 (n=1) was too small for a Kaplan-Meier Curve.

NEW COMMENT: WOULD YOU PLEASE ADD THIS TO THE LIMITATION SECTION?
